# Advances in the Study of the Mechanism by Which Selenium and Selenoproteins Boost Immunity to Prevent Food Allergies

**DOI:** 10.3390/nu14153133

**Published:** 2022-07-29

**Authors:** Kongdi Zhu, Shihui Yang, Tong Li, Xin Huang, Yulan Dong, Pengjie Wang, Jiaqiang Huang

**Affiliations:** 1Beijing Advanced Innovation Center for Food Nutrition and Human Health, Department of Nutrition and Health, China Agricultural University, Beijing 100083, China; zkd15938702692@163.com (K.Z.); shyang@cau.edu.cn (S.Y.); leetong0606@163.com (T.L.); 15501852343@163.com (X.H.); ylbcdong@163.com (Y.D.); wpj1019@cau.edu.cn (P.W.); 2Key Laboratory of Precision Nutrition and Food Quality, Department of Nutrition and Health, Ministry of Education, China Agricultural University, Beijing 100083, China; 3College of Veterinary Medicine, China Agricultural University, Beijing 100083, China

**Keywords:** selenium, selenoprotein, immune function, food allergy

## Abstract

Selenium (Se) is an essential micronutrient that functions in the body mainly in the form of selenoproteins. The selenoprotein contains 25 members in humans that exhibit a number of functions. Selenoproteins have immunomodulatory functions and can enhance the ability of immune system to regulate in a variety of ways, which can have a preventive effect on immune-related diseases. Food allergy is a specific immune response that has been increasing in number in recent years, significantly reducing the quality of life and posing a major threat to human health. In this review, we summarize the current understanding of the role of Se and selenoproteins in regulating the immune system and how dysregulation of these processes may lead to food allergies. Thus, we can explain the mechanism by which Se and selenoproteins boost immunity to prevent food allergies.

## 1. Introduction

Se is an essential trace element for humans and is an important component of selenoprotein [1]. It is an essential micronutrient that exists in the form of selenocysteine (Sec), the 21st amino acid, upon its incorporation into selenoproteins via the tRNA(Sec) encoded by Trsp, which is involved in a number of physiological processes in individuals. The biological effects of Se are mainly exerted through its incorporation into selenoproteins as Sec [2]. The research on the effects of Se and selenoproteins on the immune function of animals and humans began in the early 1970s [3]. It has been reported that Se is an important component of immune system regulation in humans and animals, mainly involved in regulating the functions of T lymphocytes, B lymphocytes, NK cells, and neutrophils. Existing studies have also shown that a mild deficiency can lead to impaired immune function, such as cell oxidation, degeneration, and damage to immune organs, resulting in reduced immunity and consequently various diseases [4]. When Se is deficient, the body’s ability to synthesize immunoglobulins decreases, and the ability of T lymphocytes to differentiate and proliferate decreases. Nutritional level of Se or a certain range of intake of Se higher than the nutritional level can enhance the immunity of individuals, improve antibody production, increase the activation and proliferation ability of B lymphocytes, enhance the function of T lymphocytes, increase the number of neutrophils, and can promote the penetration of Se into immune organs such as spleen, liver, and lymph nodes [5]. Adequate intake of selenium is essential to maintain good health. For adults, a minimum of 40 µg/day is required to support minimum expression of Se enzymes, and 300 µg/day is required to reduce the risk of cancer development [6]. However, excessive intake of Se can damage cellular and humoral immune functions. Many pathological conditions involving the immune system can be affected by the Se status in individuals.

In addition, the toxicity of Se and its compounds largely depends on their chemical form and dose. It is reported that inorganic selenium species are more toxic, and the concentration range between insufficient and excess intake is quite narrow [7]. Therefore, both the form and dose of Se intake are critical for individuals’ health, and detailed studies are needed due to the available data on toxic doses of Se [8]. Selenium and its compounds, such as selenomethionine, sodium selenite, and methyl selenocysteine, have more potent anticancer and antioxidant effects at high doses [9]. Nevertheless, the toxicity of high doses of selenite has raised concerns due to the small gap between the beneficial and toxic effects of selenite. Therefore, toxicity thresholds have been a key issue in the development of selenium supplements and biopharmaceuticals [10]. Many studies have shown that selenium nanoparticles are less toxic, more bioavailable, and more bioactive than inorganic or organic selenium-containing compounds, such as inducing selenium-containing proteins, scavenging free radicals and preventing immune-related diseases through potent antioxidant effects [11,12,13].

As the prevalence of food allergies (FAs) increases worldwide, it is particularly important to study the mechanism of food allergy and its prevention. Food allergy is an atopic immune response that can be harmful to human’s health [14]. It is commonly associated with skin, gastrointestinal, and respiratory symptoms such as atopic dermatitis, allergic rhinitis, asthma, and even shock. With the development of precision nutrition, specific nutrients can be supplemented to prevent, control, and treat diseases associated with them, such as Se supplementation to prevent food allergies and related immune diseases [15]. This article will provide an overview of the role of selenoproteins, which have immune functions, in improving immunity and thus preventing food allergies.

## 2. Selenium and Selenoproteins

Se is an essential trace element that possesses the symbol Se with atomic number 34 in the periodic table. It was discovered by Swedish chemist Jons Jacob. Se exists in two different forms, namely organic and inorganic. Organic forms of Se are present as selenocysteine and selenomethionine in the human body. Numerous experimental results have shown that Se is mainly present in the tissues and cells of the human and animal immune systems in the form of selenomethionine and selenocysteine [16]. Inorganic forms such as selenite and selenate are accumulated in plants through soil. Se was first identified as an essential element in the body of life in 1957. As one of the essential trace elements in animals, it is mainly involved in multiple life activities in the form of selenoproteins, which act as antioxidants, among other things. Se is necessary for living organisms because it is a component of the Sec found in various selenoproteins. Selenoproteins are synthesized in cells by a unique mechanism involving specific enzymes and factors. The human selenoproteome is encoded by 25 selenoprotein genes. The functions of the encoded proteins in the body are extremely diverse. Many selenoproteins have significant antioxidant activity and therefore play a crucial role in cellular antioxidant defense as well as in the maintenance of cellular redox homeostasis. This activity explains their importance in a variety of biological processes, such as cell ageing, iron death, and specially immune system activity [17]. Selenoproteins are mainly proteins containing selenocysteine, which in animals, like the other 20 known amino acids, is the product of a strict UGA codon-mediated translation and is therefore known as the 21st amino acid. However, in animals, UGA is usually present as a stop codon, and therefore, a selenocysteine insertion sequence (SECIS) is required to co-direct the encoding with UGA. The process is initiated by the binding of serine (Ser) to the Sec-specific tRNA:tRNASec. The serine residues of Sec-tRNASec are enzymatically phosphorylated and generated with the Se donor Se monophosphate (Se~P) to produce Sec-tRNASec. Sec-tRNASec is delivered to the ribosome mediated by the selenocysteine elongation factor (EFSec). SECIS-binding protein 2 (SBP2) links SECIS to the ribosome and facilitates this transfer, and together with EFSec and SECIS, SBP2 forms the SECIS complex [18].

Twenty-five selenoproteins have been identified in humans, twenty-four of which are present in rodents in a Sec-containing form [19], highlighting the value of rodent models in determining the role of members of the selenoprotein family in the immune response. Selenoproteins are widely distributed in the tissues of animals and also perform a wide range of functions [2]. While most selenoproteins function as selenases involved in redox reactions, some do not, and the functions of these non-enzymatic members are gradually being explored.

## 3. Effects of Selenoproteins on the Immune System

Se is present in a variety of the major immunological organs in vivo such as bone marrow, thymus, liver, spleen, and lymph nodes. At the cellular level, Se is found in lymphocytes, granulocytes, and monocyte/macrophages [4]. Some selenoproteins are involved in cellular activation and differentiation and are important for innate and adaptive immune responses [20]. The most complete selenoproteases associated with immune function include glutathione peroxidases (*GPXs*), thioredoxin reductases (*TXNRDs*), iodothyronine deiodinases (*DIOs*), methionine-R-sulfoxide reductase B1 (*MSRB1*), and selenite synthase 2 (*SPS2*). For non-enzymatic selenoproteins, the most characteristic one in terms of immune cell function is selenoprotein K (*SELENOK*) [21]. Table 1 lists several selenoproteins that have been demonstrated in animal models to have a large impact on the immune system and their specific functions, and the roles of these selenoproteins in different immune cells and tissues are discussed in more detail in the following section.

**Table 1 nutrients-14-03133-t001:** Immune function of related selenoproteins.

Selenoprotein	Abbreviation(s)	Immune Function	Species	References
Selenoprotein K	*SELENOK*	Promote effective Ca^2+^ flux during immune cell activation	Mice	[13,22,23,24]
Selenoprotein S	*SELENOS*, *SEPS1*	Retrograde translocation of misfolded proteins from the ER to regulating ER stress; modify immune-cell activity	Mice	[25]
Thioredoxin reductase 1	*TXNRD1, TrxR1, TR1*	Act as a regulator and a regulated target in the macrophage gene expression. Maintain redox tone in immune cells through regeneration of reduced cytosolic TXN1	Mice	[26]
Cytosolic glutathione Peroxidase	*GPX1*	Highly expressed in macrophages and T cells as an antioxidant. Important in T-cell activation and differentiation	Mice	[27,28]
Phospholipid hydroperoxide glutathione peroxidase	*GPX4*	Highly expressed in macrophages as an antioxidant	Mice	[4,29]
Deiodinases	*DIOs*	Active thyroid hormone levels may affect systemic Se available for selenoprotein synthesis in tissues involved in the immune response	Mice	[4,30]
Selenoprotein P	*SELENOP, SEPP*	Intracellular antioxidant in phagocytes; affect the differentiation of macrophages; modulate immune cell function	Mice	[31,32]
Selenoprotein H	*SELENOH*	Transcriptional regulation	Mice	[33]
Selenoprotein I	*SELENOI*	Phospholipid synthesis Act as an ethanolamine phosphotransferase affecting T-cell activation	Mice	[4,34]
Selenoprotein F	*SELENOF, SEP15*	Protein-folding; facilitates antigen presentation via ER-to-Golgi transport	Mice	[34,35]
Selenoprotein R	*MSRB1*	Promote DC-mediated T-cell priming and Th1 differentiation	Mice	[36,37]
Selenophosphate Synthetase	*SPS2*	The biosynthesis of selenoproteins	Mice	[38]
Selenoprotein T	*SELENOT*	Strongly correlated with immune-related cytokines, especially IL-1β and IL-2	Chicken	[39,40]
Selenoprotein W	*SELENOW*	Expressed in the immune system, but the regulatory mechanism is unknown	Chicken	[31,41,42]

## 4. Glutathione Peroxidases

*GPXs* are the most typical Se-containing enzymes, and the main role of *GPX* is to promote hydroperoxide metabolism and reduce damage to the body. *GPXs* use their active center of Sec to eliminate reactive oxygen species (ROS), including hydrogen peroxide and phospholipid hydroperoxides. There are eight isoforms of human *GPXs*, of which *GPX1–6* are selenocysteine enzymes [20]. The first selenoprotein to be identified in mammals was *GPX1*. As the most abundant selenoprotein in mammals, *GPX1* is an enzyme found in the cytoplasm and mitochondria and catalyzes the reduction of GSH-dependent H_2_O_2_ to water. *GPX2* is found mainly in the epithelium of the gastrointestinal tract, while *GPX3* is excreted mainly from the kidneys and is the predominant form of *GPX* in plasma. *GPX4* is expressed in a variety of cell types and tissues, whereas *GPX6* is only found in the olfactory epithelium and during embryonic development [43]. *GPX1* and *GPX4* are expressed in most tissues but also in some immune tissues and cells. The highest-expressed selenoprotein mRNA in both macrophages and T cells is *GPX1*, which acts as an antioxidant. Meanwhile, *GPX4* is also highly expressed in macrophages [27]. A recent study by Hoffmann et al. showed a higher and more sustained oxidative burst during activation of CD4+ T cells in *GPX1*^-/-^ mice compared to wild-type controls, and an another in-depth study revealed a specific role for *GPX1* in T-cell activation and differentiation [28].

## 5. Thioredoxin Reductases

*TXNRDs* are subfamily of another selenoprotein family. *TXNRD1-3* are Se-containing flavinases, which contain the selenocysteine residue (SeCys). Their main function is to reduce small molecular proteins thioredoxin (*TRX*) and disulfide isomerase. *TXNRD1* is a kind of cytoplasmic enzyme, which locates in the cytoplasm and nucleus, reduces *TXN1*, and is particularly important for maintaining the redox of immune cells [26]. *GPX* and *TXNRD* play complementary roles where equilibrium is a key factor in the modulation of the immune response [20]. Macrophages are tissue-resident phagocytes derived from bone marrow and are central to the immune surveillance system. A study by Bradley et al. found that mice exposed to LPS produced more of the selenoprotein *TXNRD1* in their macrophages [44]. The findings suggest that *TXNRD1*, a selenoprotein induced by LPS, is essential for VSIG4 gene expression in resting macrophages and has a potential role in the regulation of immune responses. These results reveal that *TXNRD1* is both a regulator and a regulated target in the macrophage gene expression network and suggest a link between selenium metabolism and immune signaling. Joshua et al. reported a novel metabolic function of Sec-expressing mammalian *TXNRD* and found that *TXNRD* is one of the important mechanisms of mammalian host defense, inhibiting pathogens while limiting host tissue damage [45]. In conclusion, the current studies all indicate a critical link between TXNRD and the immune system.

## 6. Deiodinases

Deiodinases include *DIO1*, *DIO2*, and *DIO3*. Deiodinases are catalytic enzymes for the conversion of thyroid metabolism from *T4* to the active form of *T3*, of which Se is the active component [4]. The tetraiodothyronine (*T4*) hormone prodrug is activated to triiodothyronine (*T3*) or converted to trans-thyroxine by *DIO1* or *DIO2*. Thyroxine secreted by the thyroid gland is predominantly *T4*. However, the smaller amount of *T3* is the real active thyroxine, which is five to eight times more active than *T4*. Se deficiency reduces the activity of deiodinase, which alters thyroid hormone metabolism, manifesting itself as an impaired conversion of *T4* to *T3*. These three deiodinases are expressed in many tissues but rarely in immune cells. Thyroid hormone levels affect selenoprotein synthesis in tissues throughout the body, so deiodinases indirectly affect the immune response [30]. There is now evidence from observational studies and randomized controlled trials that Se may reduce thyroid peroxidase antibody concentrations and hypothyroidism in the form of selenoproteins and also suggests that the appropriate status of deiodinases is critical for thyroid health [46].

## 7. The Thioredoxin-Like Rdx Family

Selenoprotein T (*SELENOT*) is a highly conserved enzyme with a thioredoxin-like fold localized to the ER membrane [47]. It is involved in redox regulation, and low expression of *SELENOT* leads to oxidative stress and immune dysfunction [48]. A strong correlation between *SELENOT* mRNA expression and immune-related cytokines, especially IL-1β and IL-2, was found in those immune organs by correlation analysis, suggesting that the immune response is more sensitive to *SELENOT* mRNA expression. The high correlation between *SELENOT* mRNA expression and immune-related cytokines may be due to the sensitivity of Se content to Se deficiency [39]. In immune tissues, *SELENOT* and synthesis-related factors were also sensitive to Se content. In a recent study by Pan et al., *SELENOT* mRNA expression levels, immune function, and antioxidant function were suppressed in broiler immune organs due to Se deficiency. Dietary Se deficiency inhibits *SELENOT* expression and the ability of *SELENOT* to regulate oxidative stress while impairing the multi-effective mechanisms of the immune response [40]. Thus, it is also possible to identify the important role *SELENOT* plays in immune function.

Another selenoprotein of the thioredoxin-like Rdx family is Selenoprotein W (*SELENOW*). Selenoprotein W (*SELENOW*) is expressed in the immune system of mammals [31]. A current study suggests that *SELENOW* may play an important role in the protection of avian immune organs from inflammatory damage through the regulation of inflammation-related genes [41]. To date, the mechanisms of induction and regulation of specific *SELENOW* activation among inflammation-related genes are unclear and require further study. The research of Sun et al. suggests that Se regulates the differentiation and maturation of chicken dendritic cells (DCs) through selenoproteins, which are closely associated with surface markers of chicken DCs. Among them, SelW showed the highest correlation with the cell surface markers MHC II and CD11c [49].

## 8. Other Selenoproteins

Members of the selenoprotein family are defined by bound Sec residues, but how different selenoproteins utilize Sec functionally is quite different. Some of the biological functions include transcriptional regulation (*SELENOH*), phospholipid synthesis (*SELENOI*), protein folding (*SELENOM* and *SELENOF*), methionine thioredoxin reduction (*SELENOR*), and selenoprotein biosynthesis (*SELENOP2*). Most of these functions are required for the normal function of most tissues and cell types, including those involved in the immune response. The functions of some members of the selenoprotein family are not known or unknown. Below is an overview of several selenoproteins with more obvious immune functions based on available studies.

*SELENOP* is one of the more studied selenoproteins and the only selenoprotein with multiple Sec, which has an important role for the organism in selenium transport and redox. *SELENOP* is an extracellular glycoprotein found in almost all tissues. *SELENOP* performs important antioxidant functions and is particularly important for certain immune functions. Macrophage differentiation is also influenced by redox tone, and it has been shown that *SELENOP* plays a particularly important role in this process [17]. *SELENOP* also acts as an intracellular antioxidant in phagocytes. *SELENOP* has also been shown to affect macrophage migration, possibly by increasing the expression of matrix-associated genes [32]. However, the exact role of *SELENOP* in the immune cell population remains rather obscure, and much work is needed to thoroughly elucidate the microenvironment-dependent role of Se and *SELENOP* [31]. 

Similar to *SELENOP*, methionine sulfoxide reductase B1 (*MSRB1*) acts as a oxidoreductase and also acts as a redox agent in the body. The effect of *MSRB1* on the immune system has been studied mainly in the function of Th1. *MSRB1* is also a selenoprotein that contains Sec residues that catalyze redox reactions. *MSRB1* has been found to be important for human health, but how *MSRB1* shapes immunity remains poorly understood [50]. *MSRB1* was found to be an antigen-presenting cell in DCs and appears to have a facilitative role in DC-mediated T-cell initiation and Th1 differentiation [36]. In the immune response, immune activation is associated with the production of intracellular ROS. In order to stimulate an effective immune response, intracellular levels of reactive oxygen species must be balanced. The current studies have shown that reactive oxygen species-mediated signaling can be regulated by *MSRB1,* and *MSRB1* plays a crucial role in the ability of DCs to provide the antigen presentation and co-stimulation required for the initiation of CD4+ T cells [51]. *MSRB1* promotes Th1 differentiation by activating the STAT6 pathway in DCs, which induces DC maturation and IL-12 production. In addition, *MSRB1* promotes follicular helper T-cell (Tfh) differentiation [31]. Upon further investigation, *MSRB1* is crucial in regulating the innate immunity response through the redox control of actin. Macrophages utilize this redox control during cellular activation by stimulating *MSRB1* expression and activity as a part of innate immunity. Thus, it was identified that the regulatory role of *MSRB1* as a antagonist in orchestrating actin dynamics and macrophage function [52]. These findings suggest a potential role for *MSRB1* selenoprotein in the innate control of adaptive immunity [53].

There are also some selenoproteins that are not oxidoreductases, and *SELENOI* is one of them. It is an ethanolamine phosphotransferase involved in the synthesis of two different ethanolamine phospholipids [37]. It is unclear how *SELENOI* may participate in phospholipid reprogramming to regulate T-cell activation and proliferation. Recent research data presented demonstrate that T-cell activation leads to increased levels of *SELENOI* enzyme. T-cell activation in *SELENOI* knockout resulted in an accumulation of ATP and decreased AMPK activation, disrupted metabolic reprograming, and reduced cell cycle progression and pro-growth pathways. These data provide new insight into how activated T cells rely on increased *SELENOI*-dependent PE and plasmenyl PE synthesis, which play an important role in metabolic reprogramming required for cellular proliferation.

*SELENOK* and *SELENOS* are two endoplasmic reticulum (ER) transmembrane proteins that are immune-related selenoproteins. These two proteins play an important role in ER stress protection of cells [21]. *SELENOK* has been shown to be important for endoplasmic reticulum stress and calcium-dependent signaling [23]. Recent studies have shown that *SELENOK* promotes Ca2^+^ flux in immune cell activation, and the molecular mechanisms of calcium dependence have been elucidated using knockout mice and transgenic cell lines [22,24]. It was found that *SELENOK* interacts with DHHC6, an enzyme in the ER membrane, and the *SELENOK*/*DHHC6* complex catalyzes the transfer of acyl groups to cysteine residues in the target protein [54]. The calcium channel protein IP3R is acylated by this process and is the means by which tetrameric calcium channels in the endoplasmic reticulum membrane are stabilized [55]. Reducing *SELENOK* levels impairs *IP3R*-driven calcium flux, and this role of *SELENOK* is important for the activation and proliferation of immune cells. *SELENOK* was crucial for promoting store-operated calcium entry (SOCE) in macrophages, T cells, and neutrophils stimulated with chemokines that trigger G-coupled protein receptor signaling in these immune cells [22]. Impaired SOCE in *SELENOK*-knockout immune cells leads to 50% reduction in activation levels and the immunity of *SELENOK*-knockout mice reduced compared to controls [22,55]. Therefore, *SELENOK* is vital for the stable expression of IP3R in ER membranes and is essential for the regulation of SOCE through such a calcium channel protein. Selenoprotein 15 (*SELENOF*) is also an endoplasmic reticulum-associated selenoprotein. It was identified in human T cells in the 1990s. *SELENOF* expression is regulated by dietary Se and is expressed in several tissues. *SELENOF* was found to be present in a UDP-glucose complex: glycoprotein glucosyltransferase (UGTR), which is involved in quality control of protein folding and is localized to the endoplasmic reticulum of cells. Ferguson et al. showed that *SELENOF* may function in the reduction of the glycoprotein substrate of UGTR or in the isomerization of disulfide bonds [56].

Through the current studies on the immunological role of selenoproteins, the mechanism of action of the relevant selenoproteins in immune cells is shown in Figure 1. *SELENOK* and *SELENOS* mainly play a regulatory role in the endoplasmic reticulum homeostasis of immune cells, thus playing a role in activating immune cells and increasing immune cell activity, with *SELENOK* also having an important role in maintaining intracellular calcium ion flux. *SELENOP*, *TXNRD1*, *GPX1,* and *GPX4* mainly act as antioxidants in immune cells and maintain the dynamic intracellular redox balance. Other selenoproteins such as selenoprotein P2 and *DIOs* influence the normal synthesis of selenoproteins. Still others such as *SELENOW* are expressed within immune cells, but the exact mechanism is not elucidated.

## 9. The Role of Selenoproteins in the Prevention of Food Allergies

### 9.1. Food Allergy and the Immune System

The basic functions of the immune system are mainly in the areas of immune defense, immune self-stabilization and tolerance, and immune surveillance. Of these, abnormalities in immune tolerance and intolerance to foreign substances are the main immunological basis for the development of allergic reactions in the body. The immunological mechanisms of food allergic reactions, as with other allergies, are currently divided into two simple categories: IgE-mediated and non-IgE-mediated. In the case of food allergy, the abnormal reactions of the body caused by the consumption of food are collectively referred to as hypersensitivity reactions. One category is non-immune-mediated and is referred to as non-allergic hypersensitivity reactions, such as those caused by enzyme deficiencies in the body or by factors such as pharmacological and toxic effects of food; the other category is immune-mediated and is referred to as allergic hypersensitivity reactions [57].

### 9.2. Ig-E-Mediated Food Allergy

The mechanisms of IgE-mediated food allergy are well understood, with IgE binding mainly to mast cells and basophils and being distributed in the skin and submucosa [58]. Its inflammatory molecules include a variety of molecules such as histamine, and its effector cells include mast cells and basophils. IgE is produced by B lymphocytes, and this function is influenced by cytokines produced by Th2 cells, mainly represented by IL4. The intracellular signaling proteins that control the secretion of cytokines such as IL4 by Th2 cells include intracellular signaling molecules such as GATA3, STAT3, and STAT6.

IgE-mediated food allergy involves sensitization and the inability to develop oral tolerance to food antigens. The mechanism of immune oral tolerance has not been elucidated, but it is clear that it may be disrupted in multiple stages. When enzymatic disruption of conformational epitopes in potential food allergens is reduced, epithelial cells secrete inflammatory cytokines such as IL-25 and IL-33 to inhibit IL-12 production [5,59]. In this condition, DCs and other antigen-presenting cells are activated and transformed into a functional pro-inflammatory phenotype. Subsequently, these antigen-presenting cells transform naive T cells into differentiated Th2 cells, which drive the expansion of allergic effector cells in B cells, such as eosinophils and mast cells [60]. It also blocks immunosuppressive functions through the release of IL-4. IgE secreted by B cells rapidly binds to the high-affinity IgE receptor present mainly on the surface of mast cells and basophils. When allergens bind to food allergen-specific IgE on the surface of these cells, their subsequent cross-linking of bound IgE triggers an intracellular cascade that leads to the release of mediators such as histamine, leukotrienes, chemokines, and other cytokines, resulting in a series of inflammatory responses. Thereafter, allergic inflammation can be maintained in the later stages of the allergic response due to the production of leukotrienes, platelet-activating factors, and cytokines such as IL-4, IL-5, and IL-13 [5]. Understanding the mechanisms of IgE-mediated food allergy can help in implementing measures to restore immune tolerance.

Selenium is able to influence allergic responses by acting on immune cells. Se may modulate food-allergic responses in already allergic individuals by affecting mast cell activity [5]. In a cell-based model, Se treatment reduced mast cell degranulation as measured by reduced release of mast cell mediators PGD2, β-hexosaminidase, and histamine [61]. In an experiment using soy as an allergen in mice, Se was found to influence sensitization to soy protein possibly by affecting DC function and T-cell differentiation. It has also been shown that Se supplementation inhibits B-cell activation, differentiation, and maturation [62].

### 9.3. Non-IgE-Mediated Food Allergy

The mechanisms of non-IgE-mediated food allergic reactions are largely not well-understood. Among these, the eosinophil-mediated mechanism is somewhat similar to that of IgE-mediated in that they both secrete cytokines via Th2 to cause further reactions, but here, IL-5 is predominantly secreted, which results in proliferation and differentiation of eosinophils, causing eosinophilic inflammation. Another type of non-IgE-mediated allergic reaction is caused by the direct release of inflammatory factors from T lymphocytes. In contrast, it has not been proven whether IgG (except IgG4) plays a role in allergic reactions.

The immunological mechanisms of food allergy, as with other allergies, are still largely unknown, especially for non-IgE-mediated allergic reactions. A better understanding of the immunological mechanisms of allergic diseases can only benefit the prevention and treatment efforts of allergic diseases.

### 9.4. Se and Selenoproteins for Immunity

Se acts in the body mainly in the form of selenoproteins, which enhance the regulation of the human immune system in several ways. Immunity is an aspect of human health that is influenced by Se levels and selenoprotein expression in the human body [63]. Se affects the body’s immunity in three main ways, namely cellular, humoral, and non-specific immunity. Se promotes the production of interferon and increases the activity of gamma-interferon in vitro, enhancing the cytotoxic effect of human NK cells without damaging target cell membranes. Se also significantly enhances the secretion of IL-1 and IL-2 from lymphocytes, stimulates the formation of immunoglobulins, and improves the body’s ability to synthesize antibodies such as IgG and IgM [64]. In addition, Se has different effects on the chemotaxis, phagocytosis, and killing of virion by phagocytes. Previous studies have shown that Se can regulate mouse helper T-cell (Th) differentiation, antibody production, leukocyte adhesion and migration, and macrophage phagocytosis [4,21]. Recent studies have confirmed that Se can regulate the immune response of DCs in chickens, mice, and humans [49,65,66,67]. Potential mechanisms by which Se is involved in the regulation of immune cell function include reactive oxygen species production and species, calcium and redox signaling, extracellular signal-regulated kinase (ERK), hypoxia-inducible factor 1α (HIF-1α), and NF-κB pathways [67,68,69,70,71,72]. A recent study in mice demonstrated that low and high Se affect the secretion of cytokines and impair immune function in the spleen [73]. It was also found that the mRNA levels of GPXs, TXNRDs, and DIOs were high in the immune organs of broiler chickens. Thus, the antioxidant selenoproteins may have an important position in the immune organs [74].

Among the 25 human selenoproteins, some have important cellular functions in antioxidant defense, cell signaling, redox homeostasis, and immune responses. Many cells are regulated by changes in redox status, usually involving the glutathione and thioredoxin systems. ROS can alter the redox status of cells, which seems to play an important role in the pathogenesis of allergic diseases. At homeostasis, the balanced production of ROS has a positive effect on combating invading pathogens. During inflammation, many phagocytes rely on ROS production to prevent damage to host cells [75]. Once this balance is disturbed, ROS levels increase dramatically, thus inducing damage to host cells [5]. Selenoproteins play an important role in maintaining the homeostatic level of ROS to protect immune cells from damage [76]. Protecting immune cells from damage caused by reactive oxygen species is essential for the proper functioning of the body’s immune system, so selenoproteins can enhance immunity by maintaining redox homeostasis in the immune system, thereby reducing inflammatory symptoms in the body and preventing food allergies. The metabolism of selenoproteins in vivo is closely related to immunity, and 0.25–0.50 mg/kg of sodium selenite was effective in enhancing antibody levels in broiler chickens with necrotizing enteritis [77]. In addition, Sun et al. found that in a lipopolysaccharide (LPS)-induced inflammation model in pigs, GPX activity showed a significant decrease, and Se metabolism was disturbed, demonstrating that the immune status of the organism can have an impact on Se metabolism [78].

Thus, Se is one of the essential trace elements to strengthen the immune system and improve the immunity of human body, and it can improve the antioxidant capacity of animals, maintain the integrity of biofilm structure, promote the humoral immunity of animals, improve the level of antibodies, enhance the cellular immunity of animals, promote the proliferation of lymphocytes, and play a vital role in the immune system of the body.

### 9.5. Reasonable Se Supplementation to Prevent Food Allergy

In the human body, Se must be in balance to prevent diseases caused by Se deficiency or excess Se. Numerous studies have shown that the incidence of food allergy is closely related to genes, the environment, and the interaction between the two [79,80]. Since genetic genes do not change significantly in a short period of time, one’s dietary factor is one of the most important reasons for the increased risk of food allergy. 

Se levels may influence oxidative stress or increased inflammation. In a mouse model of allergic asthma, dietary Se levels were associated with the development of allergic responses in mice, with high levels of dietary Se leading to higher Th2- or Th1-type immunity [81], suggesting that Se may modulate allergic responses by influencing adaptive immune responses. In an observational study of healthy children, mean plasma Se concentrations were found to be reduced by approximately 20 mg/L compared to healthy children, suggesting that children with food allergies are at higher risk of Se deficiency [82]. These data may simply point to an observable correlation of selenium levels in patients with food allergies, but there is a lack of human data relevant to the investigation of a causal relationship between Se and food allergies. 

Increasing dietary Se intake enhances Ca^2+^ flux and downstream cell signaling in CD4+ T cells, thereby significantly affecting their activation, proliferation, and differentiation [83]. The importance of adequate dietary selenium levels and its effective binding to selenoproteins for immunity has been demonstrated in cell culture models, rodent models, livestock and poultry studies, and human studies [8]. In the study of Ivory et al., Se supplementation (100 μg/day) increased plasma Se concentrations and increased T-cell proliferation and percentage of total T cells in adult human subjects [84]. Therefore, it is important to maintain the balance of Se in the human body, which can be achieved through scientific supplementation of Se and rational diet to obtain the corresponding Se content. Maintaining the balance of Se can improve the immunity of the body and prevent the development of food allergies and other related diseases. According to the FAO/WHO dietary recommendations, the intake of Se should comprise 34 μg/day for men and 26 μg/day for women. At the same time, Se produces primarily the antioxidant and immunity enhancing effects as well as certain anticarcinogenic effects at a dose of 150–200 μg/day [17,85]. The Food and Nutrition Association of the National Academy of Sciences recommends a daily intake of 50–200 μg of Se for people over 10 years of age, and the current clinically recommended safe dose is 5 μg/kg·day. Under normal circumstances, the normal physiological requirement of Se for adults is 30–400 μg/day, but long-term overdose can lead to adverse effects such as hair loss and skin darkening. The maximum allowable consumption of selenium is 300 to 600 µg/day, and the toxic dose is 900 µg/day. The damage of excessive selenium intake on the organism has been found in animal experiments. Wang et al. found that in selenium supplementation at 15 mg/kg, high selenium showed typical inflammatory features in chickens and indicated that selenium toxicity could promote inflammation through the NF-κb pathway with impaired immune function and altered Th1/Th2 balance in the spleen [86]. In recent years, approaches have emerged to treat immune-related diseases using selenium nanoparticles that are doped with various active compounds to increase their effectiveness and have fewer cytotoxic effects on healthy cells [8]. Unlike organic and inorganic selenium, nanoselenium may become the new trend of selenium supplements in the future.

In life, fish, lobster, and some crustaceans are Se-rich food; animal organs such as heart, liver, and kidney are also rich in Se; vegetables such as asparagus, peas, onions, and tomatoes also contain a certain amount of Se; garlic in spices and nuts such as Brazil nuts are particularly rich in Se. In addition, Se is also found in mineral water. Therefore, in the course of daily diet, a varied diet should be taken to effectively prevent Se deficiency and to better strengthen the body’s own immunity in order to prevent the occurrence of diseases.

## 10. Summary and Perspectives

Se is one of the essential trace elements for strengthening the human immune system and resistance to infection and functions in the organism mainly in the form of selenoproteins. It is involved in the regulation of oxidative stress, redox mechanisms, and other crucial cellular processes involved in innate and adaptive immune response through its corporation into selenoproteins [20]. As the main form of Se action in the organism, selenoproteins play an important role in the human immune system by improving the antioxidant capacity of animals, maintaining the structural integrity of biological membranes, promoting humoral immunity, increasing antibody levels, enhancing cellular immunity, and promoting lymphocyte proliferation. Food allergy is an allergic reaction caused by an immune imbalance, and its pathogenesis is closely related to the homeostasis of the immune system. Although there is still no effective treatment, it may be possible to prevent the occurrence of this allergic reaction by maintaining immune function and improving immunity due to its association with immune function. In this review, we describe the involvement of selenium and selenoproteins in several processes of the immune system that are critical for maintaining immune homeostasis and improving immunity. These may contribute to the prevention of food allergies, but further studies are still needed to understand the exact mechanisms. In addition, the safety of selenium supplementation should be considered to avoid overdose-induced selenium toxicity, and caution should be exercised in translating experimental data from animals to the human situation. Selenium interventions may be an interesting new approach for future treatment strategies for food allergy and help to improve the quality of life of food allergy patients. 

According to previous studies, nanoselenium has antibacterial, antioxidant, and anticancer activities in humans and is less toxic than organic and inorganic selenium, among which biological nanoselenium has higher biocompatibility. Therefore, nanoselenium also has a promising future as a preventive and chemotherapeutic agent for immune-related diseases. However, the mechanism of action of nanoselenium on immunity is also not fully understood, and one of the existing hypotheses is that nanoselenium may exert immune effects by enhancing oxidative stress, carcinogen detoxification, and immune surveillance [87]. Based on this paper, the immune mechanism of selenoprotein can be further investigated in the future.

## Figures and Tables

**Figure 1 nutrients-14-03133-f001:**
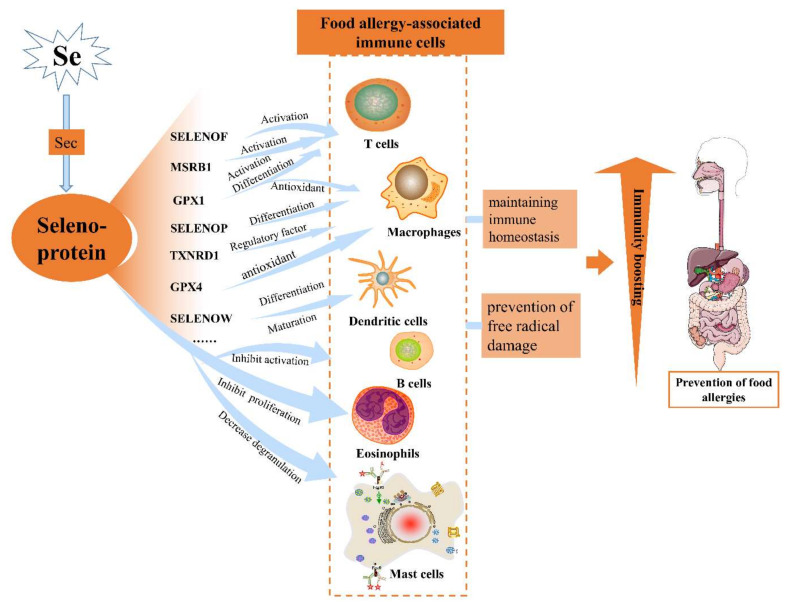
The mechanism of selenoprotein interaction with immune cells.

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
