# Peer review of "Advances in the Study of the Mechanism by Which Selenium and Selenoproteins Boost Immunity to Prevent Food Allergies"

_nutrients, 2022, doi:10.3390/nu14153133_

Round 1

Reviewer 1 Report

1. Effects of selenoproteins on the immune system. Text font must be uniform

2. Figure 1 needs an expanded legend

3. The toxicity of selenium and its compounds should be highlighted. https://pubmed.ncbi.nlm.nih.gov/34205571/

4. The Thioredoxin reductases section should be substantially expanded.

5. The Other selenoproteins section needs a schema. Should write more related to the topic of the review

6. The effects of selenium nanoparticles on immunity should be highlighted. https://pubmed.ncbi.nlm.nih.gov/35216476/

6. 

Author Response

Reply to Reviewer #1

We would like to appreciate greatly your comments that are quite helpful to improve our manuscript. We have been replying to your comments as described below.

Additional Comments:

Thank you for your appropriate evaluation to our manuscript.

Comments:

  1. Effects of selenoproteins on the immune system. Text font must be uniform.

Answer:Thank you for your correction, we have aligned the font of the "Effects of selenoproteins on the immune system" section.

  1. Figure 1 needs an expanded legend.

Answer: According to your suggestion, we collated the literature and made a supplementary drawing to Figure 1.

  1. The toxicity of selenium and its compounds should be highlighted. https://pubmed.ncbi.nlm.nih.gov/34205571/.

Answer: For the toxicity of selenium and its compounds, we have been supplemented in the sections “Introduction” and “ Reasonable Se supplementation to prevent food allergy” by consulting the recommended literature. In addition, the supplement emphasizes that reasonable selenium intake is beneficial to human health and the harm caused by excessive selenium, indicating the toxicity of selenium and its compounds.

In details, we add the descriptions “ The toxicity of Se and its compounds largely depends on their chemical form and dose. It is reported that inorganic selenium species are more toxic, and the concentration range between insufficient and excess intake is quite narrow. Therefore, both the form and dose of Se intake are critical for individuals’ health and detailed studies are needed due to the available data on toxic doses of Se. Se and its compounds, such as selenomethionine, sodium selenite and methyl selenocysteine, have more potent anticancer, antioxidant effects at high doses. Nevertheless, the toxicity of high doses of selenite has raised concerns due to the small gap between the beneficial and toxic effects of selenite. Therefore, toxicity thresholds have been a key issue in the development of selenium supplements and biopharmaceuticals. Many studies have shown that selenium nanoparticles are less toxic, more bioavailable and more bioactive than inorganic or organic selenium-containing compounds, such as inducing selenium-containing proteins, scavenging free radicals and preventing immune-related diseases through potent antioxidant effects.” in the Introduction section and “The maximum allowable consumption of selenium is 300 to 600 µg/day and the toxic dose is 900 µg/day. ” in the section of “ Reasonable Se supplementation to prevent food allergy”.

  1. The Thioredoxin reductases section should be substantially expanded.

Answer: The section on thioredoxin reductase was not sufficiently summarized in the literature reviewed at the time of writing. Based on your suggestion, we have reorganized the current experimental research literature on thioredoxin reductase and immunity and have added this section to the description.

We add the descriptions “Macrophages are tissue-resident phagocytes derived from bone marrow and are central to the immune surveillance system.A study by Bradley et al. found that mice exposed to LPS produced more of the selenoprotein TXNRD1 n their macrophages. The findings suggest that TXNRD1, a selenoprotein induced by LPS, is essential for VSIG4 gene expression in resting macrophages and has a potential role in the regulation of immune responses. These results reveal that TXNRD1 is both a regulator and a regulated target in the macrophage gene expression network and suggest a link between selenium metabolism and immune signaling. Joshua et al reported a novel metabolic function of Sec-expressing mammalian H-TrxR and found that TrxR is one of the important mechanisms of mammalian host defense, inhibiting pathogens while limiting host tissue damage. In conclusion, the current studies all indicate a critical link between TXNRD and the immune system.” in the thioredoxin reductases section.

  1. The Other selenoproteins section needs a schema. Should write more related to the topic of the review.

Answer: The other selenoproteins section includes a number of immune-related selenoproteins reported in the literature so far, which are classified into the other selenoproteins section in addition to the previously described selenoproteins with clear classifications, i.e., selenoprotein species without proven more definite effects or less studied. According to your suggestion, we have organized and summarized some links of these selenoproteins and made changes to this section to make the section more organized.

We make the following modifications:For the other selenoproteins, the order of discussion is first adjusted to integrate the related selenoproteins for illustration. We start with the more intensively studied and more specific selenoprotein P to illustrate, we add “ SELENOP is one of the more studied selenoproteins and the only selenoprotein with multiple Sec, which has an important role for the organism in selenium transport and redox. ”and then introduce MSRB1 by“Similar to SELENOP, Methionine sulfoxide reductase B1 (MSRB1) acts as a oxidoreductase and also acts as a redox agent in the body. The effect of MSRB1 on the immune system has been studied mainly in the function of Th1.”Next, we discuss the selenoprotein with the same redox function in the immune system. We then proceed with “There are also some selenoproteins that are not oxidoreductases, SELENOI is one of them. It is an ethanolamine phosphotransferase involved in the synthesis of two different ethanolamine phospholipids ” to introduce selenoprotein I, which has no redox function. In the end, we discuss selenoprotein K, selenoprotein S and selenoprotein 15, which are related to endoplasmic reticulum pressure. We combine these three proteins in our discussion and add“ Selenoprotein 15 (SELENOF) is also an endoplasmic reticulum-associated selenoprotein. It was identified in human T cells in 1990s”. At the same time, we have also revised other parts of the text according to your suggestions.

  1. The effects of selenium nanoparticles on immunity should be highlighted. https://pubmed.ncbi.nlm.nih.gov/35216476/

Answer: According to your suggestions, we carefully reviewed the articles you recommended and read the relevant literature on nano selenium and the immune system. We found that nano selenium has strong anticancer, antioxidant and high biological activity, and plays an important role in the immune system. There are a large number of experimental research reports. This paper mainly hopes to explore the preventive effect of selenoprotein on food allergy through its effect on the immune system, so the effect of selenium nanoparticles on immune related diseases in this paper is insufficient. We believe that the role of nano selenium in the immune system can be the direction of our in-depth study and research in the next stage, Therefore, the outlook is also made in the "Summary and perspectives" part. In addition, we also made supplementary explanations about the effects of selenium nanoparticles on the immune system in the "Introduction" and "Reasonable Se supplementation to prevent food allergy" sections according to the literature.

In details, we add the descriptions “ Many studies have shown that selenium nanoparticles are less toxic, more bioavailable and more bioactive than inorganic or organic selenium-containing compounds, such as inducing selenium-containing proteins, scavenging free radicals and preventing immune-related diseases through potent antioxidant effects” in “Introduction" section, “ In recent years, approaches have emerged to treat immune-related diseases using selenium nanoparticles that are doped with various active compounds to increase their effectiveness and have less cytotoxic effects on healthy cells. Unlike organic and inorganic selenium, nanoselenium may become the new trend of selenium supplements in the future.” in "Reasonable Se supplementation to prevent food allergy" section and “According to previous studies, nanoselenium has antibacterial, antioxidant and anticancer activities in humans and is less toxic than organic and inorganic selenium, among which, biological nanoselenium has higher biocompatibility, therefore, nanoselenium also has a promising future as a preventive and chemotherapeutic agent for immune-related diseases. However, the mechanism of action of nanoselenium on immunity is also not fully understood, and one of the existing hypotheses is that nanoselenium may exert immune effects by enhancing oxidative stress, carcinogen detoxification and immune surveillance. Based on this paper, the immune mechanism of selenoprotein can be further investigated in the future.”in the "Summary and perspectives" section.

Author Response

Manuscript ID: nutrients-1807276

Reply to Reviewer #2

    We would like to appreciate greatly your comments that are quite helpful to improve our manuscript. We have been replying to your comments as described below.

Additional Comments:

Thank you for your appropriate evaluation to our manuscript.

Comments:

Thank you for submitting this manuscript to Nutrients. The manuscript aimed to review the mechanism of selenoproteins in immune response and to suggest that selenium boost immunity to prevent food allergies. It has detailed how most known selenoproteins impacted the immune system at the cellular levels. However, the manuscript needs to strengthen the relation between selenoproteins and its involvement in food allergy development and prevention. Most cited literature were based on animal studies, which does not translate to one of the suggestions claimed in this manuscript –“Se supplementation prevent the development of food allergy”. This manuscript may be better framed as on explaining advanced mechanism on how selenium is involved in immune responses at cellular levels.

The introduction section has provided some good background information on selenium, however, it would be useful to explain why food allergy is of the interest, for example, prevalence of food allergies, type of food allergies, and possible links to the immune system. The manuscript includes a nice table on the immune function of related selenoproteins from animal studies and extensive descriptions on the mechanism of selenoprotein interaction with immune cells. However, the discussion on the role of selenoproteins in preventing food allergies were insufficient to address one of the review objectives as “how dysregulation of the processes may lead to food allergies”. Human studies on selenium, immunity and food allergies can be included to review the advancement in this field. Although dietary recommendations on selenium intake was briefly mentioned, both adequacy and excessive intakes in relation to food allergies should be discussed, before moving to suggest selenium supplementations as a treatment method. I have included some specific comments:

  • Lines 37-42 – it is unclear what was the viewpoint and needs to provide supporting literature. Do authors refer to optimal selenium intake for supporting immunity? Are there any different levels of evidence support the levels of selenium intake or status indicators in promoting immunity?

Answer: Yes, we do. Since it has been shown that Se is an important component of the regulation of the immune system in humans and animals, in Lines 37-42, we describe the effects of appropriate levels of selenium intake on the immune system in terms of immune cells, immune active substances, etc., as opposed to the immune hazards associated with selenium deficiency.

There are some studies about different levels of evidence support the levels of selenium intake or status indicators in promoting immunity. According to your suggestion, we have reviewed the currently available articles on the effects of dietary selenium intake on the organism and have added to this section.

In details, we add the descriptions and some supporting literature in “Introduction" section, “Adequate intake of selenium is essential to maintain good health. For adults, a minimum of 40 µg/day is required to support minimum expression of Se enzymes and 300 µg/day is required to reduce the risk of cancer development. ”and a more detailed discussion of optimal selenium for immunity is found in “Reasonable Se supplementation to prevent food allergy” section, like the description “At the same time, Se produces primarily the antioxidant and immunity enhancing effects, as well as certain anticarcinogenic effect at a dose of 150-200 μg/day”.At the same time, we have added supporting documentation in the corresponding section.  

  • Figure 1 – good idea to provide a visual overall summary, however, arrows only refer to T cells, Macrophages, and Dendritic cells, would be useful to provide specific major routes used by selenoproteins to regulate food allergy associated immune cells, such as B cells, Mast cells and Eosinophils. The connection between immune cells and “immunity boosting” is not quite clear. Also somewhere in the text, food allergy-associated immune cells need to be explained with supporting evidence.

Answer: According to your suggestion, we collated the literature and made a supplementary drawing to Figure 1. In the meantime, we also add a description of the immune cells associated with food allergy.

We add the descriptions“Selenium is able to influence allergic responses by acting on immune cells. se may modulate food allergic responses in already allergic individuals by affecting mast cell activity. In a cell-based model, Se treatment reduced mast cell degranulation, as measured by reduced release of mast cell mediators PGD2, β-hexosaminidase and histamine. In an experiment using soy as an allergen in mice, Se was found to influence sensitization to soy protein possibly by affecting DC function and T cell differentiation. It has also been shown that Se supplementation inhibits B-cell activation, differentiation and maturation”. And we also add to the mechanism of food allergy in the section “ Food allergy and the immune system”.

  • Lines 268 – 275 – could expand to highlight the links between the immune cells response, expected association with allergy, and potential role for allergy prevention.

Answer:The writing in this section is indeed lacking, and based on your suggestion, we have added to this section by reviewing the literature, emphasizing the immune cell response and its connection to food allergy, mainly in terms of immune tolerance.

We add the descriptions “IgE-mediated food allergy involves sensitization and the inability to develop oral tolerance to food antigens. The mechanism of immune oral tolerance has not been elucidated, but it is clear that it may be disrupted in multiple stages. When enzymatic disruption of conformational epitopes in potential food allergens is reduced, epithelial cells secrete inflammatory cytokines such as IL-25 and IL-33 to inhibit IL-12 production. In this condition, DCs and other antigen-presenting cells are activated and transformed into a functional pro-inflammatory phenotype. Subsequently, these antigen-presenting cells transform naive T cells into differentiated Th2 cells, which drive the expansion of allergic effector cells in B cells, such as eosinophils and mast cells . It also blocks immunosuppressive functions through the release of IL-4. IgE secreted by B cells rapidly binds to the high-affinity IgE receptor present mainly on the surface of mast cells and basophils. When allergens bind to food allergen-specific IgE on the surface of these cells, their subsequent cross-linking of bound IgE triggers an intracellular cascade that leads to the release of mediators such as histamine, leukotrienes, chemokines and other cytokines, resulting in a series of inflammatory responses. Thereafter, allergic inflammation can be maintained in the later stages of the allergic response due to the production of leukotrienes, platelet-activating factors and cytokines such as IL-4, IL-5 and IL-13. Understanding the mechanisms of IgE-mediated food allergy can help in implementing measures to restore immune tolerance.”

  • Lines 290-292 – repetition from previous sections, should focus on explaining selenoprotein for immunity as the subheading suggests.

Answer: The section does contain repetitions, and we have trimmed the repetitions. Based on your suggestion and also based on sorting out the content of the full text, we have revised the subheading to“Se and Selenoprotein for Immunity”.

We modify “Se is one of the essential nutrients identified by the World Health Organisation and acts in the body mainly in the form of selenoproteins, which can enhance the regulation of the human's immune system in a number of ways. The immune system is one aspect of human health that is impacted by dietary Se level and selenoprotein expression ” to “Se acts in the body mainly in the form of selenoproteins, which enhance the regulation of the human immune system in several ways. Immunity is an aspect of human health that is influenced by Se levels and selenoprotein expression in the human body”In addition, we have also made some appropriate adjustments to this section to make the article more relevant to the main idea.

  • Lines 339-346 – provides a list of dietary selenium intake recommendations, need to better connect with other part of the discussions. How do they relate to food allergies? How would inadequate intake of selenium contribute to food allergies? What evidence do the current literature find in prevalence of Se deficiency or Se excess? This paragraph needs to address why selenium supplementation is suggested to prevent food allergy.

Answer:According to the previous description “In the human body, Se must be in balance to prevent diseases caused by Se deficiency or excess Se. Numerous studies have shown that the incidence of food allergy is closely related to genes, the environment and the interaction between the two. Since genet-ic genes do not change significantly in a short period of time, one’s dietary factor is one of the most important reasons for the increased risk of food allergy. ” we find some studies have shown that dietary factor is one of the most important reasons for the increased risk of food allergy.Therefore, this section introduces the discussion of dietary selenium.

We have compiled the relevant literature on the effects of inadequate intake of selenium on food allergies. We add the description “ Se levels may influence oxidative stress or increased inflammation. In a mouse model of allergic asthma, dietary Se levels were associated with the development of allergic responses in mice, with high levels of dietary Se leading to higher Th2 or Th1 type immunity. suggesting that Se may modulate allergic responses by influencing adaptive immune responses. In an observational study of healthy children, mean plasma Se concentrations were found to be reduced by approximately 20 mg/L compared to healthy children, suggesting that children with food allergies are at higher risk of Se deficiency. These data may simply point to an observable correlation of selenium levels in patients with food allergies, but there is a lack of human data relevant to the investigation of a causal relationship between Se and food allergies.”

We also add the description “ The importance of adequate dietary selenium levels and its effective binding to selenoproteins for immunity has been demonstrated in cell culture models, rodent models, livestock and poultry studies, and human studies. In the study of Ivory et al., Se supplementation (100 ug/day) increased plasma Se concentrations and increased T cell proliferation and percentage of total T cells in adult human subjects” to address why selenium supplementation is suggested to prevent food allergy.

  • The section on “selenoprotein for immunity” has quite several repetitions from previous sections on selenoproteins. A section on how selenoproteins modulates immune cells in developing food allergies would be useful.

Answer:

According to your suggestion, we reorganized the article carefully and after discussion decided to revised the subheading to“Se and Selenoprotein for Immunity” to make the next revision.For this section, we discuss the effects of selenium and selenoprotein on immunity according to their respective effects, and compile the current representative relevant experimental studies on selenium and selenoprotein to improve immunity as a support to illustrate. For the role of selenoprotein and immunity we have made some additions.

We add the description “Among the 25 human selenoproteins, some have important cellular functions in antioxidant defense, cell signaling, redox homeostasis, and immune responses. Many cells are regulated by changes in redox status, usually involving the glutathione and thioredoxin systems. ROS can alter the redox status of cells, which seems to play an important role in the pathogenesis of allergic diseases. At homeostasis, the balanced production of ROS has a positive effect on combating invading pathogens. During inflammation, many phagocytes rely on ROS production to prevent damage to host cells . Once this balance is disturbed, ROS levels increase dramatically, thus inducing damage to host cells. Selenoproteins play an important role in maintaining the homeostatic level of ROS to protect immune cells from damage.Protecting immune cells from damage caused by reactive oxygen species is essential for the proper functioning of the body's immune system, so selenoproteins can enhance immunity by maintaining redox homeostasis in the immune system, thereby reducing inflammatory symptoms in the body and preventing food allergies. ”and we also modify the paragraph structure accordingly.

  • Lines 368-369 - The discussion on the pathogenesis of food allergies in relation to the immunity were insufficient, the reasoning for selenium supplementation was not discussed, therefore, the statement “We have found that Se has a beneficial effect on the immune system, so Se supplementation can be used to improve immunity and prevent food allergies” was misleading.

Answer:According to your suggestion, we find the phrase “We have found that Se has a beneficial effect on the immune system, so Se supplementation can be used to improve immunity and prevent food allergies” is indeed inappropriate and easy to misunderstand.

Thus, we delete the description “We have found that Se has a beneficial effect on the immune system, so Se supplementation can be used to improve immunity and prevent food allergies. In order to maintain the balance of Se in the organism, it is possible to obtain the appropriate amount of Se through scientific Se supplementation and reasonable diets to maintain the balance of Se in the human body in order to improve the immunity of the organism and prevent the development of food allergy and other related diseases, but the specific prevention and control mechanisms still need to be explored.” and add the description “ In this review, we describe the involvement of selenium and selenoproteins in several processes of the immune system that are critical for maintaining immune homeostasis and improving immunity. These may contribute to the prevention of food allergies, but further studies are still needed to understand the exact mechanisms. In addition, the safety of selenium supplementation should be considered to avoid overdose-induced selenium toxicity, and caution should be exercised in translating experimental data from animals to the human situation. Selenium interventions may be an interesting new approach for future treatment strategies for food allergy and help to improve the quality of life of food allergy patients.”

Reviewer 3 Report

Very interesting article

Author Response

We would like to appreciate greatly your comments that are quite helpful to improve our manuscript. We have been replying to your comments as described below.

Additional Comments:

Thank you for your appropriate evaluation to our manuscript.

Round 2

Reviewer 1 Report

The authors have done a serious work on the article. As presented, it may be accepted for publication.

Reviewer 2 Report

Dear Kong-Di and colleagues, 

Well done to revise this manuscript. The addition of detailed description and discussion together with some re-orgranisation has greatly improved the quality of this manuscript. 

Thank you.